# Study on the Location-Routing Problem in Network-Type Tractor-and-Trailer Transportation Mode

**Qingbin Wang \*, Xiaolin Liu, Gang Li and Jianfeng Zheng**

College of Transportation Engineering, Dalian Maritime University, Dalian 116026, China; liuxiaolin@dlmu.edu.cn (X.L.)
\* Correspondence: wangqingbin@dlmu.edu.cn

**Abstract:** Under the trend of developing green transportation in China, tractor-and-trailer transportation has received more attention. This paper focuses on the network-type tractor-and-trailer transportation mode in the port hinterland, aiming to tackle the problems of low efficiency and customer satisfaction in the existing transportation network. The authors recommend considering opening several alternative depots and making vehicle scheduling decisions simultaneous in order to optimize the existing transportation network. Therefore, this paper constructs a bi-level programming model with a generalized total cost minimization as the objective function. The solution to the original problem is divided into two stages: the location-allocation problem and vehicle scheduling; a two-stage hybrid heuristic algorithm is designed to solve the problem. Through the continuous iteration of the upper genetic algorithm and the lower hybrid particle swarm algorithm, the overall optimization of the problem is achieved. Finally, a specific example verifies the model and the algorithm's effectiveness. The results show that the method proposed in this paper can significantly improve customer satisfaction and reduce transportation costs to a certain extent. It can also provide effective theoretical decision support for logistics enterprises to carry out tractor-and-trailer transportation business and develop green transportation.

**Keywords:** green transportation; transport network design; location-routing problem; port hinterland; bi-level programming model; two-stage hybrid heuristic algorithm





## 1. Introduction

The transportation industry is the basic industry of national economic development. Regardless, the resulting large amount of carbon emissions inevitably brings a series of environmental problems [1] and therefore is also an important area of national energy conservation, emission reduction, and sustainable development. Tractor-and-trailer transportation is a special form of transportation in which the power section and the load section of the vehicle can be separated, which can not only improve transportation efficiency [2], but also has been proven to reduce fuel carbon emissions [3–5]. Compared with Western countries, the development of tractor-and-trailer transportation in China is relatively late and the mode is mostly one tractor towing one trailer. The development of tractor-and-trailer transportation has become one of the most important ways for China's transportation industry to turn to green, low-carbon, and sustainable development.

Network-type tractor-and-trailer transportation is a product of a particular stage of tractor-and-trailer transportation development, commonly found in the collection and distribution system in the hinterland of ports. Its characteristics are a stable supply of goods and a mature transportation network. There are one or more depots in each transportation network which dispatch and maintain vehicle resources (tractors and trailers) in the system and assume certain storage, operation, and turnover functions. Any two nodes in the system may generate trailer (heavy trailer or empty trailer) transportation demand between them. Heavy and empty trailers refer to the trailer with or without cargo inside, respectively.

The basic operation process of network-type tractor-and-trailer transportation is to transport all trailers in the system from the origination node to the corresponding destination and finally return to the depot during the decision-making period. Simplified as shown in Figure 1, the solid and dotted lines represent the transportation task paths of the heavy and empty trailers, respectively.

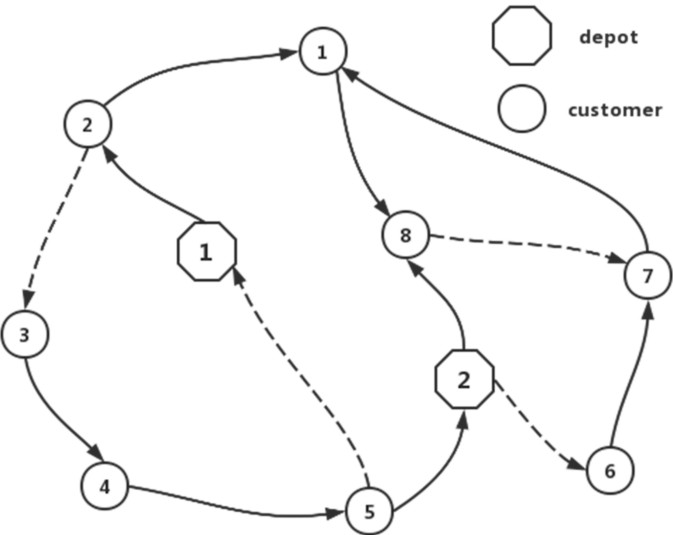

**Figure 1.** Network-type tractor-and-trailer transportation.

Different from the traditional vehicle scheduling problem, the vehicle scheduling problems of tractor-and-trailer transportation have their particularity and complexity due to their many scheduling subjects. Chao [6] first defined this problem as TTRP, established an integer programming model, and designed a tabu search algorithm to solve it. Caris et al. [7] constructed the container-hauling problem at the terminal as a vehicle scheduling problem with a time window, designed a two-stage algorithm to generate the initial solution, and accelerated convergence through three neighborhood search algorithms. Xu et al. [8] have established a model of tractor-and-trailer transportation for enterprises and designed specific transportation routes. He et al. [9] considered the factors of carbon emissions from the perspective of environmental protection and studied the problem of inland container tractor-and-trailer transportation. With the deepening of research, Bian et al. [10] considered the three forms of transportation routes of the complete vehicle, truck, and hybrid under the tractor-and-trailer transportation mode, built the model with the travel time as the objective function, and solved it using a two-stage hybrid heuristic algorithm. Yang et al. [11–13] took into account the uncertainty of the empty trailer-dispatching task, the combined transportation of different types of tractors and trailers, and the uncertainty of the trailer operation time, then specifically designed a multi-stage algorithm to solve these problems. In terms of the problem of container tractor-and-trailer transportation from port to customer, Lu et al. [4] proved the advantages of tractor-and-trailer transportation in energy conservation and emission reduction. Xue et al. [14] developed a max-min ant colony optimization algorithm to solve the problem of local container drayage. You et al. [15] established a robust dual-objective model and used the ant colony algorithm embedded within the Zoutendijk feasible direction method to solve it.

In network-type tractor-and-trailer transportation mode, the number and locations of the depots are related to the radiation range and operation efficiency of the transport network. However, the current research on the location problem of tractor-and-trailer transportation is relatively tiny. Wei et al. [16], based on the application of the improved barycenter method, determined the location of the depot from an economic perspective. Fu et al. [17] built a cold-chain logistics center-location model with minimum cost as the objective function and considered the low-carbon factor. Zhong et al. [18] set up a

mathematical model based on the tractor-and-trailer transportation mode of the inland-port freight station yard, solved it using a genetic algorithm, and finally verified the location superiority through the simulation model. Wang et al. [19] divided the location decision into two stages: first, use the fuzzy analytic hierarchy process to obtain the candidate depot and then design an integer-programming algorithm to solve it accurately.

In addition, although location and vehicle scheduling are under different decision-making frameworks, from a long-term perspective, either considering location alone or vehicle scheduling will result in reduced transportation efficiency or even increased costs. The combination optimization problem of the two is called the LRP, or the location-routing problem. In response to this problem, Ferreira et al. [20] proposed two simulated annealing algorithms based on the greedy principle to allocate clients. Farham et al. [21] added the time window factor based on this problem and proposed combining the column generation algorithm with branch pricing to solve this problem. Marinakis et al. [22] proposed an improved particle-swarm optimization algorithm considering demand uncertainty. Yanfang et al. [23] took into account the actual operation of manufacturers and cold-chain logistics companies, launched a bi-level programming model based on conflict cooperation, and designed a GAPSO hybrid optimization algorithm to solve. Li et al. [24] considered the impact of existing hubs on the transportation network and decided on an opening-and-closing hub system on this basis.

To sum up, some achievements have been made in researching the location-routing problem. However, under the mode of tractor-and-trailer transportation, most articles only consider the single-location problem or vehicle-scheduling problem and there are few integrated studies of the two. Moreover, most of the optimization research on the location-routing problem is on the subject of building a new transportation network, but in reality, the transportation network already exists in many cases. We only need to decide on opening or closing the alternative depot and vehicle-scheduling on this basis. Although opening more depots can address growing customer demand and improve customer satisfaction, it may also overlap with the radiation area of the original depot in the transportation system and even cause an increase in logistics costs. Balancing the two and making reasonable decisions is a problem that must be considered by enterprises carrying out tractor-and-trailer transportation business.

Therefore, in the mode of network-type tractor-and-trailer transportation, this paper considers the depot location and vehicle-scheduling problem as a combinatorial optimization problem, uses a mixed time window to describe customer satisfaction, and constructs a bi-level programming model with minimum generalized cost as the objective function, to obtain the optimal decision-making scheme of depot selection and vehicle scheduling.

## 2. Problem Description

Network type tractor-and-trailer transportation is commonly found in the collection and distribution systems in the hinterland of ports, where there is a stable cargo flow and a large customer base. There are one or more depots and multiple customers in the transportation system and transport demands may rise between each operation point. The depot undertakes the role of maintaining and repairing the tractors and trailers and all the tractors are parked in these depots.

Based on the existing network-type tractor-and-trailer transportation system in the port hinterland, this paper considers opening several alternative depots to jointly carry out the tractor-and-trailer transportation task with the original depot. It is known that all tractors start from the depot and return after completing the transportation tasks according to the dispatching plan. Because some transportation tasks need to retrieve goods from the depot for operation or turnover, the depot can also be considered a particular customer node. However, after the opening of the new depot, this part of the transportation task needs to be reassigned to the corresponding depot.

Under the condition that the depot and some transportation tasks are uncertain, to optimize the operation scheduling of the entire transportation system it needs to be carried out in three stages: (1) make the location decision, that is, determine the number and location of depots; (2) determine the transportation task, that is, match the depot to the transportation task to determine each task's origination node and destination node; (3) schedule vehicle planning, that is, under the condition that the depot and transportation tasks are clear, complete all transportation tasks within the decision-making period and obtain the scheme that minimizes the generalized total cost of the whole transportation system.

## 3. Bi-Level Programming Model

### 3.1. Model Assumptions

This model is based on the following assumptions: the shortest distance between any two nodes in the transportation network is known; due to the stable cargo flow of the network-type tractor-and-trailer transportation system, all transportation tasks are known and one tractor can only tow one trailer; the number of tractors and trailers is sufficient to complete all transportation tasks; the loading and unloading time of the trailer are constant and known; the average speed of the tractor is constant and known; there is no isomerism between tractor and trailer; the transportation cost of a tractor driving alone, towing an empty or heavy trailer, is linear with the travel distance, and the coefficient is known.

### 3.2. Parameters and Symbols

$N$: The set of alternative depots, $n \in \{1, 2, \ldots, N\}$

$P_0$: The existing depot in the transportation system

$f_n$: Fixed cost of the alternative depot "$n$"

$f'_n$: Variable cost of alternative depot "$n$", the operation cost of the unit transportation task

$q_n$: The number of transportation tasks assigned to an alternative depot "$n$" that need to be completed by the depot;

$Q$: The number of all transportation tasks that need to be completed by the depot;

$P$: The set of all selected depots, $P_0 \in P$

$G$: The set of all customer nodes

$V$: The set of all nodes in the transportation network, any two nodes are represented by $(i, j)$, $V = P \cup G$

$d_{ij}$: The distance from node "$i$" to node "$j$" (km), $\forall i, j \in V$

$K$: The set of tractors, $k \in \{1, 2, \ldots, K\}$

$K_p$: The set of tractors owned by the depot "$p$", $k_p \in \{1, 2, \ldots, K_p\}$

$M$: The set of all transportation tasks, $m \in \{1, 2, \ldots, M\}$

$M_k$: The set of transportation tasks of tractor "$k$", $m_k \in \{1, 2, \ldots, M_k\}$

$M_a, M_b$: The set of empty/heavy trailer transportation tasks

$O_m, D_m$: The origination/destination node of task "$m$"

$t_m, t'_m$: The start/end time of task "$m$"

$$kh = \begin{cases} \text{Tractor travelling alone,} \, h = 1 \\ \text{Towing empty trailer,} \, h = 2 \\ \text{Towing heavy trailer,} \, h = 3 \end{cases}, \forall k \in K, h \in H = \{1, 2, 3\}, \text{ three driving}$$

states of tractor

$c_0$: Fixed cost of tractor dispatching (CNY)

$c_h$: Transportation cost per unit distance driven by tractor in "$h$" state (CNY/km)

$T_1$: Maximum working time of tractor in decision-making period (h)

$T_2$: Time for the tractor to hook up/drop off a trailer (h)

$V_0$: Average speed of tractor (km/h)

$F_t$: Customer satisfaction with the original transportation system

$f(t'_m)$: Mixed time window penalty function, used to describe customer satisfaction, depends on the time when the tractor completes the task "$m$", which is defined as follows:

$$f(t'_m) = \begin{cases} +\infty, & t'_m < E_m \\ a(BestE_m - t'_m), & E_m < t'_m < BestE_m \\ 0, & BestE_m < t'_m < BestL_m \\ b(t'_m - BestL_m), & BestL_m < t'_m < L_m \\ +\infty, & t'_m > L_m \end{cases} \tag{1}$$

$[BestE_m, BestL_m]$ is the most ideal time window of task "$m$", and $[E_m, L_m]$ is the acceptable time window of task "$m$". That is to say, when the tractor completes the task between $[BestE_m, BestL_m]$, the penalty cost is 0; if the task is completed earlier or later, we need to pay a time penalty cost and the penalty coefficients are $a$ and $b$, respectively; however, if it is earlier than $E_m$ or later than $L_m$, the customer cannot accept it, so the time penalty cost is infinite.

$Z_n$: Decision variables. If alternative depot "$n$" is selected, it is 1, otherwise, it is 0

$y^w_{mk}$: Decision variables. If task "$m$" is performed by tractor "$k$" as its "$w$-th" task, it is 1, otherwise, it is 0

$x^{kh}_{ij}$: Decision variables. If tractor "$k$" passes from node "$i$" to node "$j$" in state "$h$", it is 1, otherwise it is 0

### 3.3. Mathematical Model

The upper-level model mainly considers the fixed cost of the depot and the variable cost of the depot operation, with the lowest cost as the objective function. The objective function and constraints are as follows:

$$\min F_1 = \sum_{n=1}^{N} \left( f_n + f'_n q_n \right) Z_n \tag{2}$$

$$\sum_{n=1}^{N} Z_n \geq 1 \tag{3}$$

$$\sum_{n \in P} q_n = Q \tag{4}$$

$$\sum_{m \in M} f(t'_m) < F_t \tag{5}$$

$$Z_n \in \{0, 1\}, n \in \{1, 2, \dots, N\} \tag{6}$$

Constraint (3) means that the selected depot is at least one. Constraint (4) indicates that all the depot operation tasks are assigned; Constraint (5) is a result constraint, indicating that the customer satisfaction with the new location solution should be higher than the original one. Constraint (6) is the value constraint of decision variables.

The lower-level model mainly considers the scheduling problem of the tractor, uses the mixed time window to describe customer satisfaction, and considers minimizing the generalized total cost as the decision-making goal. The objective function and constraints are as follows:

$$\min F_2 = c_0 \sum_{k \in K} \left( \frac{\sum_{m \in M_k} y^w_{mk}}{|M_k|} \right) + \sum_{k \in K} \sum_{i,j \in V} \sum_{h \in H} c_h x^{kh}_{ij} d_{ij} + \sum_{m \in M} f(t'_m) \tag{7}$$

$$\sum_{k \in K} y^w_{mk} = 1, \forall m \in M \tag{8}$$

$$x^{k2}_{O_m D_m} = 1, \forall m \in M_a \cap M_k \tag{9}$$

$$x^{k3}_{O_m D_m} = 1, \forall m \in M_b \cap M_k \tag{10}$$

$$\sum_{i \in V} \sum_{h \in H} x^{kh}_{ij} = \sum_{i \in V} \sum_{h \in H} x^{kh}_{ji}, \forall j \in V, k \in K \tag{11}$$

$$\sum_{i \in V} \sum_{h \in H} x^{kh}_{pi} = \sum_{i \in V} \sum_{h \in H} x^{kh}_{ip} = 1, \forall p \in P, k \in K_p \tag{12}$$

$$\sum_{h \in H} x^{kh}_{ij} = 0, \forall i, j \in P, k \in K \tag{13}$$

$$t'_m \leq t_{m+1}, \forall k \in K, m \in M_k \tag{14}$$

$$t'_m \leq L_m, \forall m \in M \tag{15}$$

$$t'_m \geq E_m, \forall m \in M \tag{16}$$

$$\frac{\sum_{i,j \in V} \sum_{h \in H} x^{kh}_{ij} d_{ij}}{V_0} + 2T_2 M_k \leq T_1, \forall k \in K \tag{17}$$

$$y^w_{mk}, x^{kh}_{ij} \in \{0, 1\}, \forall i, j \in V, k \in K, h \in H, m \in M, w \in M_k \tag{18}$$

Among them, the first item of the objective Function (7) corresponds to the fixed cost of starting the tractor. The second item represents the transportation cost of the tractor in the three different driving states of the tractor driving alone and towing the empty or heavy trailer, respectively. The third item represents the penalty cost of violating the most ideal time window.

Constraint (8) means that each transportation task is executed once. Constraint (9) and Constraint (10), respectively, specify the driving state of the tractor when carrying out the transportation task of the empty and heavy trailer, respectively. Constraint (11) represents the flow balance of the trailer at any node. Constraint (12) means that all tractors drive out of the depot and finally return to their depot. Constraint (13) means that the tractor cannot go directly from one depot to another. Constraint (14) indicates the task sequence constraint of the tractor. For any tractor, subsequent transportation tasks can be started only after the previous one is completed. Constraints (15) and (16) are task time-window constraints. Constraint (17) is the maximum working time constraint of the tractor. Constraint (18) is the value constraint of the decision variable.

In this bi-level programming model, the upper-level model directly affects the lower-level vehicle scheduling problem through location decisions while the lower-level model can reject some location decisions that do not meet the requirements through result Constraint (5) and continuously iterate with the upper-level model to discard solutions with unsatisfactory results until the overall optimal solution is found.

## 4. Algorithm Design

As a branch of the vehicle-scheduling problem, the vehicle-scheduling problem of tractor-and-trailer transportation has been proven to be an NP-hard problem which is extremely difficult to solve. The location-routing problem of tractor-and-trailer transportation proposed in this paper is more complex than this. Most existing relevant studies use heuristic algorithms to solve the problem. This paper uses some excellent algorithm ideas for reference, combines the genetic algorithm, clustering algorithm, and hybrid particle swarm

algorithm, and proposes a two-stage hybrid heuristic algorithm to solve the problem. The overall process framework of algorithm design is shown in Figure 2.

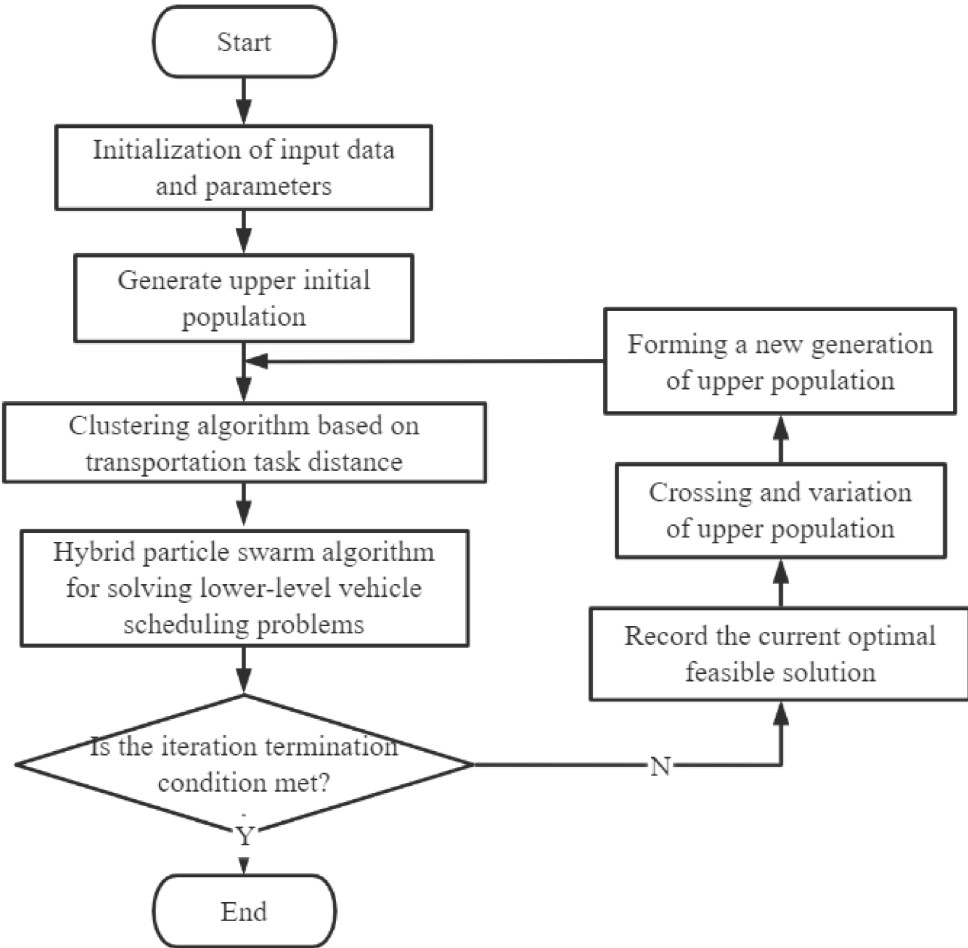

**Figure 2.** Algorithm flow chart.

For the problem of which depots to choose for the upper-level model, when the number of alternative depots is large, the evolution rules of the genetic algorithm can effectively eliminate some inferior solutions from participating in the calculation, thus reducing the scale of problem-solving. At the same time, because some tasks in the original transportation system need to be performed by the depot and after the opening of the new depots, the specific depot to complete these tasks is uncertain, which requires us to reassign the transportation tasks. A simple way to solve this problem is to cluster all tasks according to the distance between the task nodes and the depot so that all transportation tasks are determined.

On the premise that the location of the depot and the origination and destination nodes of all tasks have been determined, the lower-level model becomes a vehicle-scheduling problem with multiple depots. Although the conventional particle-swarm algorithm has a high search speed, it is easy to fall into local optimization and the results are not ideal when solving practical problems. Therefore, three neighborhood search strategies of individual particles are added to the design of the algorithm to improve the particle-swarm algorithm and its global search ability. The specific steps of the algorithm are as follows:

- Step1. The cluster method based on the distance between the task nodes and the depot is used to set the boundary and determine the corresponding relationship between each transportation task and the depot. Then the multi-depot problem is transformed into N problems with a single independent depot.
- Step2. Determine the coding rules. Adopt the integer coding method with the task as the object; each line represents the order of a vehicle performing the task.
- Step3. Generation of the initial solution. Disrupt all unexecuted tasks in the task pool and randomly add a task to the task sequence until the maximum working time limit of the tractor is reached. When adding follow-up tasks, if more than one task meets the requirements, priority should be given to adding tasks with a shorter preparation distance (i.e., no-load traveling distance) to form a better initial solution. This cycle continues until all task sequences are output to create the initial solution.
- Step4. Particle update strategy based on elite selection. Before updating particles, calculate the fitness of $n$ particles in the current population and retain the former n/2 individuals as elite individuals to participate in the iterative update. The formula for updating is as follows:

$$\begin{cases} V_i^{k+1} = wV_i^k + c_1\left(P_i^{best} - P_i^k\right) + c_2\left(P_g^{best} - P_i^k\right) \\ P_i^{k+1} = P_i^k + V_i^{k+1} \end{cases} \tag{19}$$

where, $V_i^k$ and $P_i^k$ represent the velocity vector and position vector of particle $i$ at the $k$-th cycle, $P_i^{best}$ and $P_g^{best}$ represent the particle's historical optimal solution and global optimal solution, $k$ represents the number of iterations, $w$ represents the inertia factor, and $c1$ and $c2$ represent the self-cognitive factor and social cognitive factor respectively. In the process of updating, the particles keep themselves unchanged with a certain probability close to the individual historical optimal solution and close to the global optimal solution and finally generate a new generation of n/2 particles.

- Step5. The remaining n/2 particles of the new generation are generated by three neighborhood search strategies. The original n/2 elite individuals accept one of their three transformations (exchange, insertion, 2 opt) with a certain level of probability. So far, a new generation of particles has been produced.
- Step6. Return to Step 4 and start a new iteration until the stop criteria are met.

## 5. Numerical Verification

### 5.1. Example Design

In this section, we select part of the existing transportation business of a transportation enterprise in the hinterland of the port to verify the effectiveness of the model and algorithm.

It is known that the enterprise relies on the customer group in the hinterland of the port and the demand for tractor-and-trailer transportation is relatively stable. Still, the radiation capacity of the existing depot is limited and cannot meet growing customer requirements. To improve customer satisfaction, the transportation company plans to open a new depot and cooperate with the original depot to complete the tractor-and-trailer transportation task jointly.

The information on the original depot $P_0$ and the five alternative depots $N_i(i = 1, 2, \ldots, 5)$ is shown in Table 1. The depot area is 500 m$^2$ and the rental price per square meter is about 0.67–0.83 CNY/day based on the annual service cycle. The shortest distance between each node is shown in Table 2. The transportation task information is shown in Table 3. There are 40 transportation tasks in total. The first 10 are empty-trailer transportation tasks, the last 30 are heavy-trailer transportation tasks, and the task designated as "node 0" is the transportation task that requires depot operation.

**Table 1.** Information regarding original depot and alternative depots.

| Depot | Coordinate | Cost of Using the Depot (1000 CNY/Year) | Cost of Using the Depot (CNY/Day) | Trailer Operation Cost (CNY/Each) |
|---|---|---|---|---|
| $P_0$ | (135,145) | 0 | 0 | 180 |
| $N_1$ | (128,202) | 130 | 356.2 | 180 |
| $N_2$ | (192,215) | 130 | 356.2 | 150 |
| $N_3$ | (111,61) | 150 | 411 | 180 |
| $N_4$ | (128,76) | 120 | 328.8 | 150 |
| $N_5$ | (66,239) | 140 | 383.6 | 150 |

**Table 2.** Information of original depot and alternative depots.

|  | $P_0$ | $N_1$ | $N_2$ | $N_3$ | $N_4$ | $N_5$ | 1 | 2 | 3 | 4 | 5 | 6 | 7 | 8 | 9 | 10 | 11 | 12 | 13 | 14 | 15 |
|---|---|---|---|---|---|---|---|---|---|---|---|---|---|---|---|---|---|---|---|---|---|
| $P_0$ | 0 | 57 | 90 | 87 | 69 | 117 | 116 | 41 | 21 | 159 | 102 | 135 | 67 | 132 | 70 | 132 | 134 | 181 | 140 | 89 | 83 |
| $N_1$ | 57 | 0 | 65 | 142 | 126 | 72 | 173 | 75 | 37 | 109 | 72 | 107 | 92 | 136 | 22 | 113 | 142 | 135 | 183 | 59 | 33 |
| $N_2$ | 90 | 65 | 0 | 174 | 153 | 128 | 190 | 73 | 74 | 154 | 136 | 170 | 148 | 201 | 45 | 178 | 87 | 184 | 159 | 9 | 91 |
| $N_3$ | 87 | 142 | 174 | 0 | 23 | 184 | 48 | 105 | 108 | 228 | 159 | 181 | 82 | 139 | 157 | 168 | 191 | 242 | 140 | 174 | 160 |
| $N_4$ | 69 | 126 | 153 | 23 | 0 | 174 | 50 | 83 | 90 | 218 | 152 | 178 | 81 | 144 | 139 | 167 | 168 | 235 | 123 | 154 | 147 |
| $N_5$ | 117 | 72 | 128 | 184 | 174 | 0 | 225 | 145 | 102 | 44 | 31 | 48 | 107 | 111 | 85 | 65 | 212 | 64 | 252 | 120 | 39 |
| 1 | 116 | 173 | 190 | 48 | 50 | 225 | 0 | 117 | 136 | 268 | 202 | 227 | 129 | 187 | 184 | 215 | 185 | 285 | 109 | 192 | 197 |
| 2 | 41 | 75 | 73 | 105 | 83 | 145 | 117 | 0 | 45 | 184 | 137 | 171 | 108 | 173 | 76 | 170 | 95 | 209 | 107 | 75 | 107 |
| 3 | 21 | 37 | 74 | 108 | 90 | 102 | 136 | 45 | 0 | 142 | 92 | 127 | 76 | 135 | 49 | 126 | 130 | 166 | 151 | 71 | 66 |
| 4 | 159 | 109 | 154 | 228 | 218 | 44 | 268 | 184 | 142 | 0 | 71 | 69 | 150 | 142 | 115 | 90 | 241 | 31 | 292 | 145 | 77 |
| 5 | 102 | 72 | 136 | 159 | 152 | 31 | 202 | 137 | 92 | 71 | 0 | 35 | 79 | 82 | 91 | 42 | 215 | 83 | 242 | 128 | 46 |
| 6 | 135 | 107 | 170 | 181 | 178 | 48 | 227 | 171 | 127 | 69 | 35 | 0 | 99 | 73 | 125 | 21 | 250 | 67 | 275 | 162 | 79 |
| 7 | 67 | 92 | 148 | 82 | 81 | 107 | 129 | 108 | 76 | 150 | 79 | 99 | 0 | 68 | 113 | 86 | 202 | 161 | 193 | 144 | 95 |
| 8 | 132 | 136 | 201 | 139 | 144 | 111 | 187 | 173 | 135 | 142 | 82 | 73 | 68 | 0 | 158 | 52 | 265 | 139 | 261 | 194 | 122 |
| 9 | 70 | 22 | 45 | 157 | 139 | 85 | 184 | 76 | 49 | 115 | 91 | 125 | 113 | 158 | 0 | 133 | 127 | 144 | 180 | 37 | 47 |
| 10 | 132 | 113 | 178 | 168 | 167 | 65 | 215 | 170 | 126 | 90 | 42 | 21 | 86 | 52 | 133 | 0 | 254 | 87 | 270 | 170 | 89 |
| 11 | 134 | 142 | 87 | 191 | 168 | 212 | 185 | 95 | 130 | 241 | 215 | 250 | 202 | 265 | 127 | 254 | 0 | 271 | 105 | 96 | 173 |
| 12 | 181 | 135 | 184 | 242 | 235 | 64 | 285 | 209 | 166 | 31 | 83 | 67 | 161 | 139 | 144 | 87 | 271 | 0 | 317 | 175 | 102 |
| 13 | 140 | 183 | 159 | 140 | 123 | 252 | 109 | 107 | 151 | 292 | 242 | 275 | 193 | 261 | 180 | 270 | 105 | 317 | 0 | 166 | 215 |
| 14 | 89 | 59 | 9 | 174 | 154 | 120 | 192 | 75 | 71 | 145 | 128 | 162 | 144 | 194 | 37 | 170 | 96 | 175 | 166 | 0 | 83 |
| 15 | 83 | 33 | 91 | 160 | 147 | 39 | 197 | 107 | 66 | 77 | 46 | 79 | 95 | 122 | 47 | 89 | 173 | 102 | 215 | 83 | 0 |

**Table 3.** Information of transportation task.

| Task No. | Origination/Destination Nodes | Best Time Window | Acceptable Time Window |
|---|---|---|---|
| 1 | (0,2) | [6,10] | [6,11] |
| 2 | (0,7) | [12,16] | [11,17] |
| 3 | (3,5) | [9,13] | [8,14] |
| 4 | (3,2) | [7,11] | [7,12] |
| 5 | (10,6) | [16,20] | [15,20] |
| 6 | (12,5) | [12,16] | [11,17] |
| 7 | (13,1) | [10,14] | [9,15] |
| 8 | (13,9) | [8,12] | [7,13] |
| 9 | (14,9) | [8,12] | [7,13] |
| 10 | (14,8) | [16,20] | [15,20] |
| 11 | (0,3) | [16,20] | [15,20] |
| 12 | (0,10) | [7,11] | [6,12] |
| 13 | (0,13) | [14,18] | [13,19] |
| 14 | (1,2) | [16,20] | [15,20] |
| 15 | (1,5) | [12,16] | [11,17] |
| 16 | (2,7) | [6,10] | [6,11] |
| 17 | (2,11) | [8,12] | [7,13] |
| 18 | (2,14) | [12,16] | [11,17] |
| 19 | (3,1) | [16,20] | [15,20] |
| 20 | (3,4) | [6,10] | [6,11] |

**Table 3.** *Cont.*

| Task No. | Origination/Destination Nodes | Best Time Window | Acceptable Time Window |
|---|---|---|---|
| 21 | (4,0) | [18,22] | [15,20] |
| 22 | (4,12) | [18,22] | [17,22] |
| 23 | (5,0) | [16,20] | [15,20] |
| 24 | (5,12) | [16,20] | [15,20] |
| 25 | (5,15) | [6,10] | [6,11] |
| 26 | (6,0) | [12,16] | [11,17] |
| 27 | (7,10) | [12,16] | [11,17] |
| 28 | (7,13) | [8,12] | [7,13] |
| 29 | (8,4) | [12,16] | [11,17] |
| 30 | (8,14) | [15,19] | [14,20] |
| 31 | (9,0) | [14,18] | [13,19] |
| 32 | (9,13) | [8,12] | [7,13] |
| 33 | (10,3) | [12,16] | [11,17] |
| 34 | (10,15) | [8,12] | [7,13] |
| 35 | (11,14) | [8,12] | [7,13] |
| 36 | (12,0) | [18,22] | [15,20] |
| 37 | (13,3) | [18,22] | [15,20] |
| 38 | (14,10) | [13,17] | [12,18] |
| 39 | (15,3) | [8,12] | [7,13] |
| 40 | (15,8) | [8,12] | [7,13] |

Other parameter information is as follows: the decision-making period is 24 h; the average speed of the tractor is 75 km/h; the fixed cost of starting the tractor is 300 CNY/vehicle; the maximum working time of the tractor is 12 h; the cost of the tractor traveling alone/towing empty trailer/towing heavy trailer is 1.1 CNY/km, 1.6 CNY/km and 1.85 CNY/km respectively; The average time for loading and unloading trailers is 0.3 h; The unit penalty cost for the tractor to complete the transportation task earlier or later is 100/h and 300/h.

In terms of algorithms, the maximum number of iterations of the upper-level genetic algorithm is set to 50, the population size is 32, the crossover probability and mutation are 0.9 and 0.1, respectively, and the roulette-wheel strategy is used to screen individuals.

The maximum number of iterations of the hybrid particle-swarm algorithm is set to 600, the population size is 64, the inertia factor $w$ is set to 0.6, and the self-cognition factor $c1$ and social cognition factor $c2$ are set to 2 according to experience.

### 5.2. Results and Discussion

Three different strategies are used to calculate the example: the first strategy is to maintain the original depot $P_0$'s independent operation and obtain the optimal value of the original cost under this strategy; the second strategy is to use the bi-level programming method proposed in this paper, which considers the interaction between location and vehicle-scheduling, and takes them as a whole; the third strategy is to consider the location and vehicle-scheduling problem as two independent sub-problems; that is, first use the gravity method to make location decisions, select the same number of depots as the optimal result of Strategy II, and then solve the vehicle-scheduling problem.

Based on the above data, one should solve the above three strategies, run the algorithm ten times, and choose the most ideal solution to decode. The optimal scheme obtained by the three different strategies is shown in Table 4 and the comparison of results is shown in Table 5.

From the comparison in Table 5, it can be seen that the most ideal solution strategy II is to select depot $P_0$ and $N_5$. The generalized total cost is 13,360.8 CNY, about 4.7% lower than the original strategy of 14,021.8 CNY. The tractor transportation cost is reduced by about 3.3% and the time window cost generated by customer satisfaction is significantly decreased, by about 96.2%.

**Table 4.** The optimal scheme of three strategies.

| Strategy I (Original Scheme) | | | Strategy II (Bi-Level Programming) | | | Strategy III (Gravity Method) | | |
|---|---|---|---|---|---|---|---|---|
| Tractor No. | Depot | Task Sequence | Tractor No. | Depot | Task Sequence | Tractor No. | Depot | Task Sequence |
| 1 | $P_0$ | [1,16,20,6,13,37] | 1 | $P_0$ | [1,16,28,7,15] | 1 | $P_0$ | [4,1,17,18,13,37] |
| 2 | $P_0$ | [10,30] | 2 | $P_0$ | [4,17,35,9,2,11,19,14] | 2 | $P_0$ | [2,27,19,14] |
| 3 | $P_0$ | [28,8,31,11,19,14] | 3 | $P_0$ | [3,33,18,10,30] | 3 | $P_0$ | [16,28,7,15,11] |
| 4 | $P_0$ | [12,25,40,29,24,22,36] | 4 | $P_0$ | [32,8,31,13,37] | 4 | $N_1$ | [35,9,26] |
| 5 | $P_0$ | [18,38,5,21] | 5 | $N_5$ | [12,34,39,38,5,22,36] | 5 | $N_1$ | [12,34,39,38,5,22,36] |
| 6 | $P_0$ | [4,17,35,9,3,26] | 6 | $N_5$ | [20,25,40,29,6,23] | 6 | $N_1$ | [20,25,40,29,6,23] |
| 7 | $P_0$ | [34,39,2,27,33] | 7 | $N_5$ | [27,26,24,21] | 7 | $N_1$ | [3,33,10,30] |
| 8 | $P_0$ | [32,7,15,23] | | | | 8 | $N_1$ | [32,8,31,24,21] |

**Table 5.** Comparison of results.

| | Depot | Total Cost/CNY | Number of Tractors | Time Window Cost/Customer Satisfaction |
|---|---|---|---|---|
| Strategy I | $P_0$ | 14,021.8 | 8 | 398/5 customers' most ideal time window is not met |
| Strategy II | $P_0, N_5$ | 13,360.8 | 7 | 15/1 customer's most ideal time window is not met |
| Strategy III | $P_0, N_1$ | 14,327.4 | 8 | 38.7/1 customer's most ideal time window is not met |

In the original tractor-and-trailer transportation system, when the tractor is performing some long-distance transportation tasks, due to the limitation of its maximum working time, if it wants to perform more tasks, it can only pursue the highest overall efficiency and cost at the cost of abandoning the most ideal time window of some customers. However, after optimization, we can see that the new scheme has achieved balance between the two to a certain extent—not only has the transportation cost been reduced, but customer satisfaction has also been significantly improved in terms of both the number of customers whose most ideal time window has been met and the penalty cost.

However, when using the gravity method to solve the independent sub-problems, the optimal result of strategy III is 14,327.4 CNY, which is about 2.2% higher than the original scheme. Through the analysis of the cost composition, we can find that although the time window cost generated by customer satisfaction has decreased by about 90.3%, the transportation cost of the tractor has increased by 6.4% and the overall effect is not ideal. There are three possible reasons:

(1) The particularity of tractor-and-trailer transportation. Compared with the traditional vehicle scheduling problem, which takes the customer coordinates as the transportation orientation, the decision variable of tractor-and-trailer transportation is the task itself. That is, using the coordinates of the customer to make the location decision is not accurate.

(2) The interference of the original depot in the transportation system. Since there is already a depot in the system, the use of the gravity method to determine the location will inevitably be affected by the original depot, which is one of the reasons why the traditional gravity method is not suitable for solving such problems.

(3) The relationship between location and vehicle-scheduling. The independent sub-problem does not take into account the interaction between location and vehicle scheduling, which is also one of the factors that lead to poor results.

*5.3. Algorithm Performance Analysis*

The algorithm in this paper is programmed in Python 3.7. The running memory of the computer is 16 G and the processor is AMD Ryzen 7 4800H. The convergence effect of the global optimal solution is shown in Figure 3, indicating that the algorithm converged and achieved the global optimal value after 20 generations.

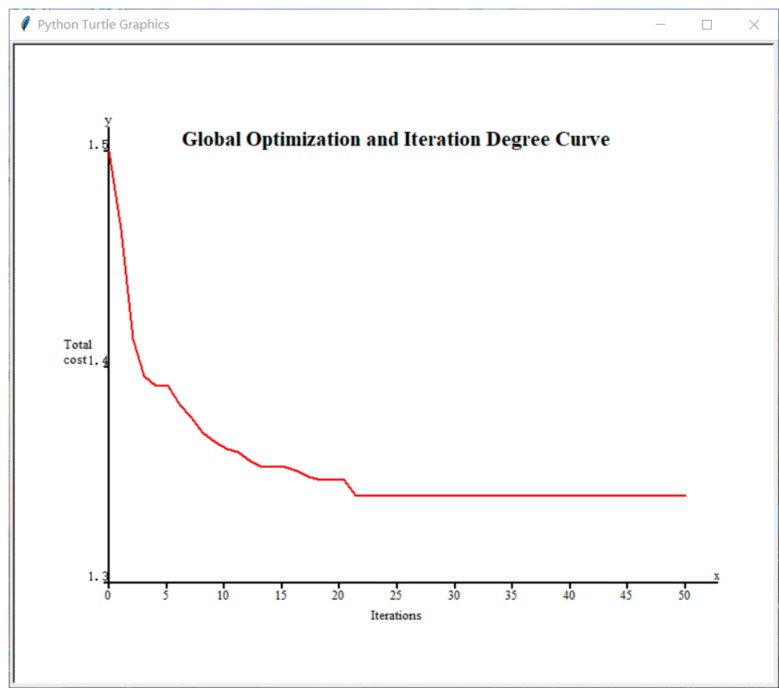

**Figure 3.** Global convergence.

When analyzing the lower layer hybrid particle-swarm algorithm independently, taking the solution of strategy II as an example, the algorithm's convergence is shown in Figure 4 and the algorithm's stability is shown in Figure 5.

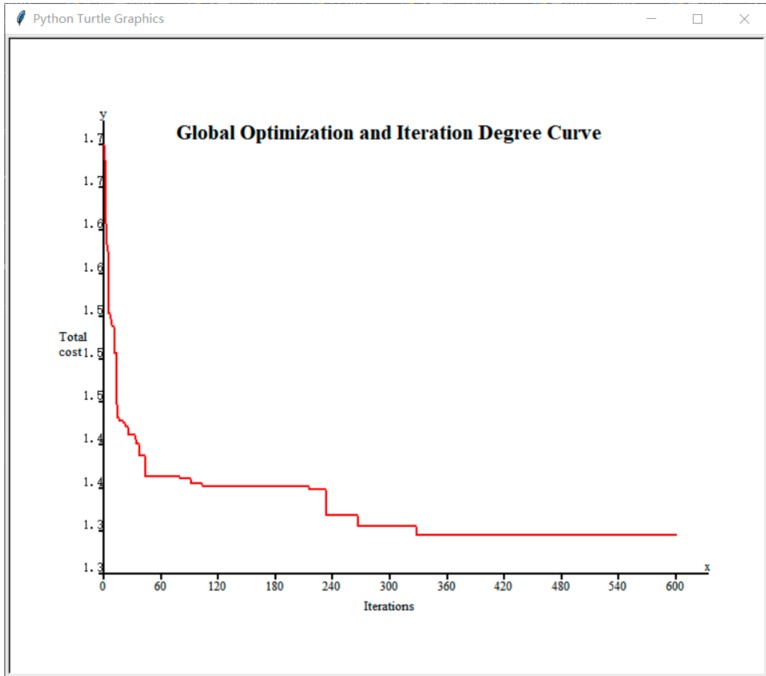

**Figure 4.** Convergence of hybrid particle swarm algorithm.

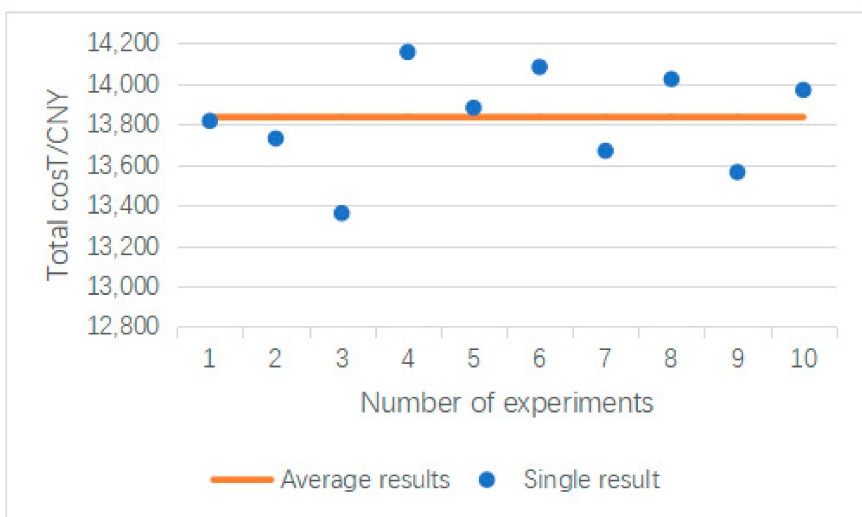

**Figure 5.** Stability of hybrid particle swarm algorithm.

We can see that the convergence rate of the objective function is faster in the first 60 generations, gradually decreases between 60 and 300 generations, and tends to be stable after 300 generations. At the same time, the algorithm can solve relatively large-scale examples in a reasonable time; as for the quality of the solution, as shown in Figure 3, the average value obtained by running the algorithm ten times with Strategy II as an example is 13,835.8 CNY, which is only 3.6% different from the optimal solution of 13,360.8 CNY. From the perspective of efficiency and stability, it can be proved that the algorithm designed in this paper is suitable for solving such problems.

## 6. Conclusions and Prospect

Under the mode of network-type tractor-and-trailer transportation in the port hinterland, aiming at the problem that the existing transportation network does not match the growing customer requirement, this paper considers opening several alternative depots to improve customer satisfaction and makes vehicle scheduling decisions at the same time. We construct a bi-level programming model with the lowest generalized total cost as the objective function. A two-stage hybrid heuristic algorithm is designed to solve the problem and a specific example verifies the effectiveness of the model and algorithm. It can provide effective theoretical decision support to develop the tractor-and-trailer transportation business and green transportation.

Through the comparative analysis of three different strategies, it can be found that although the location problem and the vehicle-scheduling problem are under different decision-making frameworks, it is necessary to consider both of them from the perspective of improving service quality and reducing transportation costs—the increase of depots in the transportation system can enhance the quality of service and significantly improve customer satisfaction, but also can reduce the total cost of tractor-and-trailer transportation to a certain extent and optimize the vehicle resource allocation of enterprises. On the other hand, the use cost of the depot is also a significant factor. When the use cost is high, although the multi-depot can improve customer satisfaction, it may also increase the logistics cost of the enterprise. How to balance the two and make reasonable decisions is a problem that needs to be considered by enterprises carrying out the tractor-and-trailer transportation business.

Finally, although rental or construction can be flexibly considered when selecting depots, the instability and variability of customers and transportation tasks still bring certain investment risks. Therefore, how to quantify the uncertain tasks and the mixed transportation of heterogeneous fleets will be one of the future research directions.

**Author Contributions:** Conceptualization, Q.W. and X.L.; methodology, Q.W. and X.L.; investigation, G.L.; resources, G.L.; data curation, G.L.; writing—original draft preparation, X.L.; writing—review and editing, Q.W. and J.Z.; supervision, Q.W. All authors have read and agreed to the published version of the manuscript.

**Funding:** This research received no external funding.

**Institutional Review Board Statement:** Not applicable.

**Informed Consent Statement:** Not applicable.

**Data Availability Statement:** All data are provided in the paper.

**Conflicts of Interest:** The authors declare no conflict of interest.

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
