# Peer review of "Study on the Location-Routing Problem in Network-Type Tractor-and-Trailer Transportation Mode"

_sustainability, doi:10.3390/su15086970_

Round 1

Reviewer 1 Report

The paper tries to optimize the existing transportation network by dividing the original problem into two stages: location-allocation problem and vehicle scheduling.

The literature review is adequate to the authors’ research.

Table 2 and Table 3 should be rearranged to fit on the page.

The motivation and conclusion for the paper are adequate.

The authors conclude from the comparative analysis of three different strategies that, it is necessary to consider both the location problem and the vehicle scheduling problem from the perspective of improving service quality and reducing transportation costs. The increase of depots in the transportation system can enhance the quality of service and improve customer satisfaction, but also can reduce the total cost of tractor-and-trailer transportation to a certain extent and optimize the vehicle resource allocation of enterprises.

Author Response

Thank you very much for your valuable suggestion!

In response to your feedback, we have made modifications to Tables 2 and 3.

Reviewer 2 Report

The submitted manuscript addresses an exciting and relevant topic in the context of green transportation. The proposed model uses an intelligent approach for optimizing routes and distributing tractor-and-trailer vehicles. The applying a genetic algorithm and hybrid particle swarm algorithm is quite justified. Despite these advantages, the paper can be improved according to the following suggestions:

1) English can be improved. It will be good to recheck the manuscript on grammar and language style, beginning from an abstract.;

2) Some keywords fully duplicate phrases from the title. It is advisable to avoid repetition in keywords. This will improve Internet searching for the paper.;

3) I was in confusion what the word 'depot' meant. If we are talking about vehicle loading location, then it is better to write 'departure point'. If denotes a cargo storage area, then it is better to write 'warehouse' or 'terminal'. Please specify additional info about this nuance.;

4) In my opinion, assumptions on a model in the subtitle of 3.1 'Model assumptions' are very simplified. Some model parameters are set as a constant. However, they are random variables (vehicle speed, cargo flow etc.) by their nature. The such statement demands an additional explanation. Because the received results can be inadequate, despite using two hybrid algorithms.;

5) In lines 145-146, it's not clear about what coefficient is talked. Explain, please.

6) I think to better understand obtained results and highlight their significance, the paragraph “Results and Discussion” can be added to the paper. This nuance will be useful for future readers.

7) Please, add the next parts: author contributions, funding etc. Please, see the template. 

Reviewer 3 Report

This paper highlights the need for integrated studies in the location-routing problem, especially for tractor-and-trailer transportation, where most articles only consider single location or vehicle scheduling problems. The optimization research on the location-routing problem mainly focuses on building new transportation networks, but in reality, the transportation network already exists in many cases. The paper emphasizes the importance of making reasonable decisions in balancing the opening or closing of depots to meet customer demand and improve satisfaction while minimizing logistics costs. My specific comments and recommendations are as follows:

1. The author claimed to optimize the LRP on the existed network. Methodologically, it is still the network design problem, regardless the existence of the network. Please clarify the novelty and necessity of the methodology.

2. Please provide more details about the specific scenario in Section 2.

3. The decision variables of upper and lower layers seem to be separate, which is not the bi-level program but a two-stage model. Please clarify the interaction between the layers.

4. The author designed a hybrid algorithm by combining two heuristic algorithms. However, there is only one convergence diagram in Figure 2. Please discuss the parameter setting and convergence of both algorithms for upper and lower layer.
